# Measuring Disability Inclusion Performance in Cities Using Disability Inclusion Evaluation Tool (DIETool)

**Nataša Rebernik** [1,2,*], **Marek Szajczyk** [3], **Alfonso Bahillo** [1] and **Barbara Goličnik Marušić** [4]

1   Faculty of Engineering, University of Deusto, 43007 Bilbao, Spain; alfonso.bahillo@deusto.es
2   Faculty of Social Sciences, University of Antwerp, 2000 Antwerp, Belgium
3   Faculty of Social Sciences, Siedlce University of Natural Sciences and Humanities, 08-110 Siedlce, Poland; marek.szajczyk@uph.edu.pl
4   Urban Planning Institute of the Republic of Slovenia, 1115 Ljubljana, Slovenia; barbara.golicnik-marusic@uirs.si
*   Correspondence: natasa.rebernik@deusto.es or natasa2.rebernik@gmail.com; Tel.: 00386-41-895-567

**Abstract:** Cities are exposed to a growing complexity, diversity and rapid socio-technical developments. One of the greatest challenges is as of how to become fully inclusive to fit the needs of all their citizens, including those with disabilities. Inclusive city, both in theory and practice, still lacks attention. Even in the context of ambitious contemporary concepts, such as smart and sustainable city, the question remains: Do smart and sustainable cities consider inclusiveness of all their inhabitants? Among numerous evaluation systems that measure city's smartness, sustainability or quality of life, those tackling inclusion are very rare. Specifically, disability inclusion is hardly covered. This may be one of the reasons why cities struggle with applying disability inclusion to practice in a holistic and integrative way. This paper proposes a Disability Inclusion Evaluation Tool (DIETool) and Disability Inclusion Performance Index (DIPI), designed to guide cities through a maze of accessibility and disability inclusion related requirements set within the political, legislative and standardization frameworks. The testing in two European cities shows that the tool is beneficial for providing diagnosis as to how disability friendly a city is, and as such offers an opportunity for designing informed corrective measures towards disability inclusive city design.

**Keywords:** inclusive city; disability inclusion; urban studies; social sustainability; smart and sustainable city; city performance metrics; Disability Inclusion Evaluation Tool (DIETool); Disability Inclusion Performance Index (DIPI)

## 1. Introduction

By UN HABITAT, an inclusive city "is a place where everyone, regardless of their economic means, gender, race, ethnicity or religion, is enabled and empowered to fully participate in the social, economic and political opportunities that cities have to offer" [1]. Modern city concepts, such as smart and sustainable city, nowadays all claim to be human-centered and inclusive. However, does this really mean that they address principles of inclusiveness for all? If so, do they incorporate evaluation, management and monitoring systems for assessing how inclusive cities are? Which of them, if any, addresses the question of disability inclusion through a set of respective indicators? Through an extensive literature review, document analysis, exchange of knowledge with scholars, practitioners and decision makers, our previous research brought a set of findings:

(1)  Cities' governments still struggle with understanding the diversity of citizens' needs.

(2)　Consequently, cities are not able to fully understand what disability inclusion means, how complex and important it is and how it can contribute to the quality of lives of a wider population of citizens, not only those with impairments.

(3)　Cities lack a holistic and systemic approach, and disability inclusive measures are still greatly of an ad-hoc nature.

(4)　Cities lack awareness and knowledge about disability inclusion, which is evidently reflected among relevant stakeholders, such as politicians, local authorities, architects, designers, planners etc.

(5)　Consequently, cities still tend to be designed in a non-inclusive manner, considering a fully abled 40-year old male as a norm for city design [2]. Hence, vulnerable groups with a set of needs that differ from such norm get excluded.

There are numerous political, legislative and standardization frameworks that may serve as guidelines. For instance, The New Urban Agenda [3], The 2030 Agenda for Sustainable Development with its goals and targets [4,5], the United Nations' paper on the right to the cities for all [6], as well as other initiatives [7–9] attempt to bring inclusive city to practice as a requirement and a norm.

Nevertheless, no exhaustive disability inclusion metrics and/or a complete guiding system has been identified that would allow cities to assess where they are in terms of disability inclusion, where their performance is low and where corrective measures need to be taken.

With this paper, we contribute to the area by outlining a novel evaluation and monitoring system as a response to the identified gap in city evaluation metrics. The herein proposed Disability Inclusion Evaluation Tool (DIETool) and its corresponding Disability Inclusion Performance Index (DIPI), in a compact and holistic way, translate the accessibility and disability inclusion requirements from the global political, legislative and standardization frameworks into a practical set of indicators at a city level. As such, they help in guiding cities towards improving their disability inclusion performance. By demonstrating how DIETool works in practice, we attempt to stimulate theoretical and empirical work towards more disability inclusive societies. We argue that smart and sustainable city initiatives should strive to become inclusive for all, and that social sustainability should encapsulate questions, such as how to design socially sustainable cities concerning social inclusion also of those with diverse impairments.

The structure of this paper is as follows: Section 2 sheds light onto the related work in the context of the identified gaps. Section 3 presents the DIETool and the DIPI main objectives, design process, structure and methodology. Section 4 demonstrates results of the DIETool testing within two European cities, Maribor (Slovenia) and Pułtusk (Poland). Finally, Section 5 compares the results, discusses the challenges faced and concludes with some research limitations and future directions.

## 2. Background

### 2.1. Smart, Sustainable and Inclusive City

Within contemporary city discussions and current global urban agendas, an inclusive city has often been referred to in broad terms of economic or social inclusion emphasizing the need for equal opportunities for all [3–6,10–15]. However, in narrow terms, inclusion reaches a greater complexity. For instance, it can be perceived on a basis of gender, age, sexual orientation, disability, religion, ethnicity, socio-cultural background, economic status, etc. For the purpose of designing fully inclusive cities, stakeholders need to understand the needs of citizens within each of these groups and make them a part of their strategic planning and implementation agendas [12,16,17]. However, cities are difficult to design and as a result, vulnerable groups with a set of needs that differ from a "norm" get often excluded [2,16–19].

Furthermore, even in the context of ambitious contemporary concepts, such as smart and sustainable city, the question remains: Do smart and sustainable cities consider inclusiveness of all their inhabitants? An extensive body of existing literature [20–28] demonstrates a significant gap and shows

that currently neither smart nor sustainable city concepts are fully ready to meet the requirements for an inclusive city as envisioned by some authors [2,9,12,14,15,29,30].

A smart city is a complex but consciously organized, purposeful system of subsystems, components and processes [26–28,31]. As such, it should make (disability) inclusion as an organic element and non-negotiable fact. The smart city concept, however, still confuses scholars, practitioners and decision makers with many different conceptualizations, "contradicting definitions and unclear measures" Having in mind such lack of cohesion, it is not surprising that it has, in theory and practice, been struggling with understanding the real meaning, importance and complexity of inclusion of diverse groups of citizens.

Sustainable city has been subjected to diverse conceptualizations, too [22,23,32], causing the need for discussions around what sustainability really means Sustainability originates from environmental studies [32–34]. Initially being concerned with sustaining balance between human well-being and nature, it has gradually grown into a conceptualization of three main dimensions: social, economic and environmental However, environmental and economic sustainability discussions have outnumbered those related to social sustainability, as the latter were introduced to sustainable development debates relatively late [35,36].

Scholars have attempted to conceptualize social sustainability, for instance, as a blend of social equity, social justice, social interaction, participation, safety, accessibility, sense of place, etc. [36–42]. Hermani et al. [43] (p. 172) see social sustainability in terms of 4'S': social capital, social cohesion, social inclusion and social equity.

According to Larimian and Sadeghi [36] (p. 5), a socially sustainable neighbourhood is "one that provides residents with equitable access to facilities, services, and affordable housing; creates a viable and safe environment for interaction and participation in community activities; and promotes sense of satisfaction and pride" among current and future generations. Despite diverse views on what social sustainability is, social equity and inclusion are part of it one way or the other. Social inclusion, viewing it holistically, entails access to environment, structures, services, processes, products and information [5–9,12,14,15,31,40], while social equity ensures that such access is fair, just and equitable to all the people (including people with disabilities). It recognizes and respects diversity and through equitable governance enables everyone to exercise their rights and duties as equal members of society [10,11,13,35,40].

This background sets the stage for further discussions around disability inclusion in the context of smart and sustainable cities for all [2,17,29,30], and questions the ways disability inclusion performance therein could be measured.

*2.2. City Evaluation and Monitoring Systems*

There is a growing presence of evaluation and monitoring systems that measure cities' attributes such as smartness [21,24,44–46], sustainability [21,36,37,47–49], accessibility [50–52], quality of life [53], creativity [54] and universal design (UD) [55].

Sustainability evaluation systems are well elaborated from environmental and economic perspectives. Where they tackle social sustainability, disability inclusion gets lost in the overall inclusion approach [36,37]. Studies have elaborated diverse metrics, for instance, in terms of economic, racial and overall inclusion [13] or social, spatial and economic inclusion However, rarely any evaluation system considers disability inclusion, such as DOTCOM tool [56], Assessment tool for inclusive cities by UNESCO [57], SDGs Advocacy toolkit [58] or at least integrates disability and accessibility related indicators as the integral part of the tool Surprisingly, even within the recent international standard ISO 37120:2018 Sustainable cities and communities–Indicators for city services and quality of life [59] within the extensive list of indicators across 19 areas, not one of the core or supplementary indicators refers to disability inclusion.

Nevertheless, disability rights are a part of the global agenda. The Standard Rules [60], United Nations Convention on the Rights of Persons with Disabilities (UNCRPD) [61], United Nations Disability

Inclusion Strategy [62], European Disability Strategy 2010–2020 [63], The European Accessibility Act [64], Americans with Disabilities Act–ADA [65] and its renewed ADA Amendments Act–ADAAA [66], and standards, such as ISO 21542:2011 for accessibility of built environment [67] provide a solid framework for ensuring that cities are also taking the path towards disability inclusive communities.

For instance, in line with these efforts, Academic Network of European Disability Experts (ANED) in collaboration with the European Commission (EC) designed an online disability evaluation tool (DOTCOM) with the aim to aggregate data for the EU member states on a country level Although the tool presents an exhaustive database for disability related policy, governance and legislation in EU member states, its approach is not suitable for measuring cities' performance in disability inclusiveness. Firstly, the tool targets countries, not cities. This unavoidably brings differences in the approach, although city governments follow national political, legislative and standardization frameworks, too. Secondly, the DOTCOM tool serves as a collection of information about countries, hence the indicators are of a generic, qualitative and broadly descriptive nature. They are not designed as performance indicators. Neither do they include a scoring methodology; thus, they do not enable cities to measure their disability inclusion performance.

Relevant disability indicators have been recently proposed within the SDGs Advocacy toolkit 27 new indicators within 19 SDGs targets have been designed and are still paving the way into the national statistical offices. Again, the indicators are directly intended for countries, not cities.

On the city level, an interesting attempt is the UNESCO's assessment tool for inclusive cities Within 11 areas of city life, it proposes between 3 and 12 simple, qualitative indicators per area. Once ready for implementation, its simplicity may in fact provide a fast performance score and help encourage cities towards considering disability inclusion. Nonetheless, its simplicity is also a drawback. It makes it incomplete, specifically in terms of diversity of disability types and heterogeneity of city's structures, services, processes, products and information that need to be covered. Finally, the tool consists of a series of closed-ended questions with no complementary indicators that would allow to understand the performance in more detail. A strong point is, however, that it envisions the collection of information on the side of city governments, as well as citizens in order to balance the obtained score.

Further on, the Access City Award [50,51] has been a joint initiative between the EC and the European Disability Forum, as an attempt to encourage cities towards accessibility. However, its aim is not a detailed evaluation of city's performance, neither in terms of benchmarking, nor in terms of a self-assessment and guidelines on how to perform better. It was designed more as an incentive for raising awareness among European cities for promoting disability inclusive practices. It is designed as a voluntary self-application process with a set of open-ended qualitative questions that may easily be answered with the main purpose of convincing the jury, rather than attempting to look in the mirror and see the real picture. As Steffan and Denninghaus [51] (p. 7) explain, one of the great challenges of such awards is, indeed, how to overcome their benchmarking nature, and among the winning cities, avoid the "window dressing" with "excuses to stop improving further".

### 2.3. City-Specific Context

With this paper, we focus on a small-scale European city context, specifically two case study cities, Maribor (Slovenia) and Pułtusk (Poland), where the pilot testing was manageable and traceable. Here, the environment is well known to the first two authors and the city administration is not too big, so that the collaboration was doable. In addition, a certain degree of commitment to the disability rights area had been proven by both cities in the past.

Both in Slovenia and Poland, strong efforts can be recognized towards establishing firm disability rights frameworks, following global and European disability rights agendas [60–64]. Beyond national and local strategies and action plans in both the countries [68–76], there was an interesting initiative developed in Slovenia to encourage cities to become disability friendly. The project of the national work-related disability association, which could be translated to "Municipality friendly to persons with disabilities" [77] assesses a candidate city based on how committed they are to ensure that inclusivity

becomes one of the main principles within their organizational structures. In fact, Maribor as our pilot case study, has been awarded with this title in 2010 after establishing a Council for Disabled People, designing an Analysis on the Rights of Persons with Disabilities [71] and drafting, at that time, the first 4-year Action Plan for Improving Equal Opportunities for Persons with Disabilities [72,73]. This was one of the first attempts to systematically and holistically approach disability inclusion in the city. Since then, the climate has positively changed among others, demonstrated also by the latest Action plan for the period 2015–2021 In the case of Maribor, the mentioned national project [77] proved to be an excellent stimulator. However, it is not very suitable as a self-assessment tool for monitoring as of how the city is performing and improving in the disability domain. The project is dependent on financial and staff capacity and it consists of an external evaluation body that evaluates the city based on an extensive descriptive application form and occasional monitoring visits.

In Poland, dedication to disability inclusion is reflected through national strategy and action plans [69,70]. An interesting Local Data Bank (LDB) [74] can also be mentioned, which is a large national statistical database that includes disability-related features. However, the data is collected at the poviat level and thus, cannot be directly applicable to a city context. Therefore, with this work, we also bridge the gap in data collection on social inclusion of persons with disabilities, giving cities an opportunity to self-assess their performance with an easy-to-use tool. Furthermore, following the global and national frameworks, also Pułtusk Poviat has shown dedication to disability inclusion by adopting a Strategy for Solving Social Problems in Pułtusk Poviat 2015–2022 [75] and Action Program for Persons with Disabilities in the Pułtusk Poviat 2015–2022 [76]. The two documents attempt to follow the progress in Pułtusk (as a poviat and the city), discuss factors influencing the inclusion of people with disabilities and help local administrations in designing future responsive measures.

## 3. Methodology and Methods

This chapter presents a novel tool (DIETool) and its associated component DIPI for self-assessment of city inclusiveness. It outlines their objectives, the design processes to achieve them, as well as the methodology of their construction and operation in a local governance context.

### 3.1. DIETool and DIPI Objectives and Design Process

The DIETool is a self-evaluation system allowing cities to measure their disability inclusion performance. It consists of 20 areas of city life (Table 1), each with a combination of carefully selected qualitative and quantitative indicators translated into a score (DIPI) that shows the city's performance against the optimum situation. The DIETool's main objectives are to offer guidance and support to local city governments. It helps identify strengths and weaknesses and provides diagnosis of disability inclusiveness in the city. As such, it can be incorporated into a holistic disability inclusive city design approach, ideally consisted of the following steps:

(1) Assessing the situation and identifying the gaps.
(2) Assessing the technical and organizational capabilities demonstrated in the area of disability inclusion performance.
(3) Discussing, designing and planning appropriate corrective measures according to the analysis.
(4) Strategic planning and action plans design.
(5) Undertaking action/Implementation of corrective measures.
(6) Creating public-private partnerships and an ongoing dialogue with citizens.
(7) Ongoing evaluation, iteration and improvement.

**Table 1.** DIETool outline with 20 areas, goals and characteristics for indicators design.

| | **Inclusive City Areas of Assessment** | | |
|---|---|---|---|
| | **Area** | **Goal** | **Characteristics/Requirements/Examples for Indicators' Design** |
| 1 | Public Spaces and Built Environment | Accessibility to public spaces and built environment. | Accessible and barrier free routes; Accessible public toilets; Accessible parks; Accessible buildings; Presence of disabled people in public; Inclusive urban planning strategies, action plans; Inclusive projects of built environment; Trained and inclusion respecting architects and designers etc. |
| 2 | Public Services and Community Supply | Accessibility to public services. | Accessible emergency number; Staff training of public servants across all services; Accessible public services institutions (post offices, banks, hospitals, medical centres, local authority offices, employment offices etc.); Sign language provision; Assistance provision; Accessible water supply; Accessible energy etc. |
| 3 | Transportation and Urban Mobility | Availability, accessibility and affordability of inclusive transportation infrastructure, products and services. | Accessible means of transportation (bus, train, tram, metro, taxi, city centre vehicles, special bicycles; Accessible information and communication about transportation; Accessible stations and platforms; Accessible ticket machines and other equipment; Accessible ticket offices, counters and other services; Available parking for disabled people; Accessible parking machines; Accessible petrol station; Accessible websites; Staff training; Funding schemes for transportation service providers; Funding schemes for special transportation equipment (e.g., adapted cars); Co-funding of transportation fees for users; Accessible information and navigation apps; Tactile guiding systems; Audio, video information support; Accessible help line; no. of disabled passengers etc. |
| 4 | Policy and Governance | Availability, accessibility and participation in policy, governance and decision-making processes. | Accessible voting procedures; Accessible voting locations; No. of disabled representatives in governance bodies; No. of bodies with disabled representatives; Participation of disabled people in strategic planning and design; Participatory governance programmes; No. of participatory projects; No. of disabled citizens participated etc. |
| 5 | Legislation and Standardization | Availability and enforcement of inclusive legislation and standardization. | Available disability legislation; Accessible disability legislation; Participation in legislation design; Enforcement of disability legislation; Available disability inclusive standards; Awareness, acceptance and enforcement of standards; Presence and incorporation of disability inclusion principles in overall legislation etc. |
| 6 | Media, Information and Communication | Accessible, democratic, inclusive and non-discriminating media, information and means of information-communication. | Inclusive, non-discriminating media contents; Awareness-raising campaigns, projects, articles about inclusion and disability; Available audio descriptions of TV and other visual media, adapted for visually impaired people; Available sign language interpretation and large captions; Accessible media websites; Staff trainings and staff trained to assist disabled people etc. |
| 7 | Education, Training and Childcare | Availability of, accessibility to and affordability of inclusive education, training and childcare for all. | Available education, training and childcare programmes, facilities, opportunities; Available funding scheme; No. of disabled people in mainstream vs. special education; No. of successful students/pupils; Available assistance services (e.g., pedagogues, personal assistance, educated teachers etc.); Available adapted literature and educational content (audio, sign language, easy to read versions); Accessible playgrounds for children etc. |
| 8 | Work, Career and Employment | Availability of and accessibility to work, career and employment opportunities, including employment rehabilitation programmes. | Available programmes for disability employment development; Funding opportunities for employers; Funding opportunities for disabled people; Accessible, inclusive employers; Accessible job posts; Accessible employers; Accessible employment offices; Accessible employment services and information; No. of disabled employees vs. unemployed and vs. non-disabled.; Success rate in job interviews; Available ergonomic solutions; Working climate and Staff training support etc. |
| 9 | Housing | Availability, accessibility and affordability of housing. | Available funding programmes and schemes; Development plans for accessible housing; No. of disabled people in accessible housing units (in terms of application, actual accommodation); Funding programmes for private owners to support inclusive and equal access to housing etc. |
| 10 | Health Care and Social Security | Availability, accessibility and affordability of inclusive health care, medical services, rehabilitation and social security programmes and schemes. | Available programmes and schemes; Financial accessibility; Accessible infrastructure; Information-communication accessibility of services; No. of accessible hospitals, medical and rehabilitation centres; Success rates of rehabilitation programmes; No. of disability specific schemes; Staff training; No. of available SL interpreters etc. |
| 11 | Assistive Technology and Independent Living | Availability, accessibility and affordability of assistive technology and ICTs. | Available programmes and schemes to support access to assistive technology, aids and support independent living; Financial access and aids; Legislation on assistive technology etc. |
| 12 | Family Life | Awareness, climate and accessibility to information, educational programmes and support towards creating family life. | Awareness raising activities; Support services; No. of disabled people with their own family, No. of disabled people in their own household; No. of disabled people with children etc. |
| 13 | Economy, Business and Industry | Accessibility to products, services and information, business opportunities and inclusive business climate. | Funding programmes for disabled business owners; Business climate; Access to incubators, business offices etc. Accessible market; Accessible services and products; Economic stability of disabled people/Financial safety/poverty index among disabled people etc. |

**Table 1.** *Cont.*

| | Inclusive City Areas of Assessment | | |
|---|---|---|---|
| | **Area** | **Goal** | **Characteristics/Requirements/Examples for Indicators' Design** |
| 14 | Finance and Financial Security | Financial safety, availability and accessibility of financial services, infrastructures and information. | Financial security among disabled people; Funding schemes for disabled people; Access to financial information; Access to financial institutions; Financial literacy trainings; Accessible ATMs; Accessible banks; Staff training in financial sector etc. |
| 15 | Community Life and Civil Initiative | Accepting non-discriminating, equality-based community climate. | Community activities, events engaging disabled people; Community climate; Acceptance levels in neighbourhoods; Accessible civil initiative programmes and actions; Fund raising for disabled people; Collaboration with NGOs etc. |
| 16 | Recreation, Sports and Leisure | Availability, accessibility and affordability of recreational activities, sports and leisure. | Accessible recreational, leisure and sports facilities; Leisure and recreational parks with adapted equipment; Available and accessible funding for sports of disabled people; Funding for talented disabled children, youngsters, sportsmen and sportswomen; Accessible events etc. |
| 17 | Culture, Arts, Cultural Heritage and Tourism | Availability, accessibility and affordability of culture, arts, cultural heritage and tourism infrastructure, items/products/works/artefacts, services, contents, information and events. | Accessible cultural, artistic and heritage programmes, contents, services, information and facilities; Funding schemes for supporting accessible culture and arts; Funding schemes to support disabled artists; Accessible cultural institutions; Accessible events; Accessible websites; Audio support and large captions for visually impaired; Sign language use in theatres, films, shows, events, celebrations; Affordable fees for disabled people; Accessible toilets at events outdoors; Staff training and staff trained to assist disabled people; Available services of assistance for disabled people etc. |
| 18 | Religion | Availability and accessibility of inclusive religious facilities and rituals. | Accessible religious institutions; Accessible and adapted rituals; Accessible websites and information; Accessible religious events; Staff training and staff trained to assist disabled people etc. |
| 19 | Technology and Innovation | Availability, accessibility and the use of inclusive technology and support to innovation | Accessibility to, availability and ownership of digital technology among persons with disabilities; Levels of digital literacy among persons with disabilities; Funding programmes for the development of accessible technology, research and innovation; Incentives for innovation among persons with disabilities etc. |
| 20 | Safety, Quality of Life and Independent Living | Ensuring safety and quality of life of persons with disabilities. | Ensuring physical and psychological safety among persons with disabilities; Domestic and non-domestic violence rates; Safe public spaces; Safe transportation; SOS system with Safe houses for women and children with disabilities; Satisfaction levels with the quality of life among persons with disabilities etc. |

A city is a system of subsystems that need to work together. In this sense, the areas within the DIETool serve as subsystems to be explored for designing an inclusive city. They complement one another and, in a whole, represent an organic blend within urban life. Under each of the areas, the DIETool identifies main goals to be pursued. Aligned with the goals, the tool then outlines a list of more detailed characteristics and/or requirements towards reaching disability inclusion. Taking an example of area 3, the goal is to achieve "accessible transportation", which can be done through an extensive list of actions for ensuring accessible means of transportation, accessible facilities (e.g., stations, platforms, information offices, ticketing machines), accessible transportation related information and services, all in forms suitable for physically, sensory and intellectually impaired people. For the purpose of evaluating disability inclusion performance of a city, we also shed light onto strategic approach, staff training, funding availability, safety, etc., in the respective area. These characteristics (as seen in Table 1) act as specific goals and serve as guidelines for further indicators for design.

The DIETool has been developed through several phases as follows:

In phase 1, we identified the 20 areas for assessment and to each of them assigned main goals.

In phase 2, a list of characteristics/requirements (as more specific goals) has been prepared for each of the respective areas. Whereas the main goals define an overall availability, accessibility and affordability of the areas, characteristics and specific goals go deeper into exploring services, products, facilities, programmes, training, contents, information and events, including the main requirements for their accessibility. With the list of characteristics, we tried to address diversity and complexity of citizens' needs, having in mind the challenges that diverse groups of disabled people face while living in a city.

Phase 3 brought a list of indicators, each consisting of:

(1) Indicator's definition (based on the identified disability rights documents with their policy guidelines, legal requirements and accessibility standards).
(2) Methodology definition (the unit, the calculation method and scoring scale definition).
(3) Data sources identification (e.g., open sources, specific city departments, other potential sources).

Phase 4 focuses on definition of overall indexing methodology and a scoring system for the DIPI.

Phase 5, finally envisions a complementary repository of relevant disability rights documents, guidelines, standards, data collection techniques (questionnaires, indirect data search, repository search, software design to support data collection, etc.) and protocol design to help stakeholders in implementing the DIETool analysis and improving their disability inclusion performance.

The selection process of the 20 areas (Table 1) builds on a careful consideration of disability inclusion and disability rights literature, such as UNCRPD [61], European Disability Strategy 2010–2020 [62], Standard Rules on the Equalization of Opportunities for Persons with Disabilities of the United Nations [60], and the European Accessibility [64]. Act These documents emphasize a vast list of areas, where the rights of persons with disabilities need to be protected. For instance, the right to life, freedom of speech, equal recognition before law, liberty and security, freedom from violence and abuse, participation in life, equal opportunities, the right to living independently, the right to accessibility (e.g., built environment, information, services and products), personal mobility, home and family, accessible education, work and employment, financial security, adequate housing, social protection, participation in political life, leisure and sports, access to culture and tourism, religion and more. Based on this extensive list, the mentioned smart and sustainable city resources gave additional tips on how to compact this vast array of disability rights' areas into a list of 20 DIETool areas, in which full enjoyment of city life should be ensured also for persons with disabilities.

The DIETool indicators were then designed, which required: (a) a detailed analysis of accessibility and inclusion principles and characteristics within each of the areas based on the disability rights legal requirements and accessibility standards; (b) exploration of what could potentially be available data resources and feasible ways of data collection in the city; (c) methodology definition and scoring scales design for each indicator; (d) rethinking the weighting system and an overall DIPI methodology

design. Before deciding on whether the indicator should be a part of the DIETool or not, we created a list of desired criteria. Ideally, in the following criteria each indicator should fulfil:

(1) Data refers to the city level (not e.g., individual, corporate, region or country level).
(2) Data collection is feasible (data exists, is available, can be retrieved, collected or gathered from existing databases, running activities or proposed DIETool data collection techniques).
(3) Data can be objectively measured and compared, can be estimated based on available information, or explored through participatory user-centered techniques.
(4) Data as the main observed attribute reflects either input (e.g., disability inclusive strategy, available funding schemes, % of annual funding available for accessible transportation) or output of city's inclusiveness (e.g., % of accessible buses, % of passengers with disabilities, % employed persons with disabilities, satisfaction levels).

It should be noted that even though it may be difficult to obtain some data, the DIETool points out its importance, thus a city is encouraged to find a way and do so. The DIETool can also guide the city in this process by proposing alternative data collection techniques and as an ongoingly upgrading repository of main political, legislative and standardization documents.

### 3.2. DIETool and DIPI Methodology

The DIETool uses a scoring methodology. As seen from the tables of the selected areas, available in the supplementary material (Tables S1–S9), each indicator is given a number, an indicator name, description, methodology description, a unit and a scoring scale. Several different scoring scales have been built, depending on the nature of the indicator, the unit used for calculation (e.g., %, no.), known reference scoring scales, etc. In all cases, the scoring scales range from 0 to 5, where 0 corresponds with extremely poor performance and value 5 for excellent performance. The overall performance score for each area is given by the normalized average score of all the indicators calculated within the respective area. The total DIPI score for the city is then calculated by the normalized average score of all the areas.

The DIETool is designed holistically and thus aiming at obtaining the overall performance index DIPI across all the areas. However, it can also be used to measure each area separately (e.g., by separate city departments). As such, it serves as a monitoring tool in order to identify the gaps and discrepancies from the optimum state in that specific working area. As a tool, it is organized in an easy way by providing ready Excel templates with indicators as seen in Supplementary Materials, Tables S1–S9. In a separate spreadsheet, it automatically calculates the scoring and provides the graphic presentation of the results as seen in Figures 1–4. With some initial guidance, it can be used independently by city officials for evaluation and analysis, future planning and monitoring.

It must be said, however, that the tool, at this point, does not include an elaborated weighting system or a benchmarking approach. When trying to correspond with this requirement, we decided to get the experience from practice first. A weighting methodology design is a complex process, that may be greatly context dependent (e.g., city-size, city's economic status, etc.). Currently, the DIETool offers a framework for the preliminary DIPI score in order to demonstrate that a city's disability inclusion performance can be measured according to the proposed legal requirements. As such, the DIETool with its methodology, is ready to be used by any city, considering possible local, regional and national specifics, but not yet ready for comparison between different cities. A thorough consideration of weighting methodology, as well as possible diversity of contexts for applying the tool is a matter of its future methodological upgrade.

### 3.3. Case Studies Definition and Approach

The DIETool was tested in real urban environments of two European cities with the aim to demonstrate how it can be used in practice and how lessons learned can feed its future development. The criteria for selection were three: Firstly, we needed an easy access to the city government. The selected two cities (Maribor, a medium-sized Slovenian city with app. 95,000 inhabitants, and Pułtusk,

a small Polish town with app. 19,500 inhabitants) are hometowns of the first two authors of this paper, who in the past already worked closely with the respective city councils. This greatly helped in accomplishing the criteria. Secondly, we needed the city mayors to say "yes", which they did without hesitation. And thirdly, we needed them to ensure enough resources in terms of time and staff capacity. They did, by assigning to the project a responsible coordinating department.

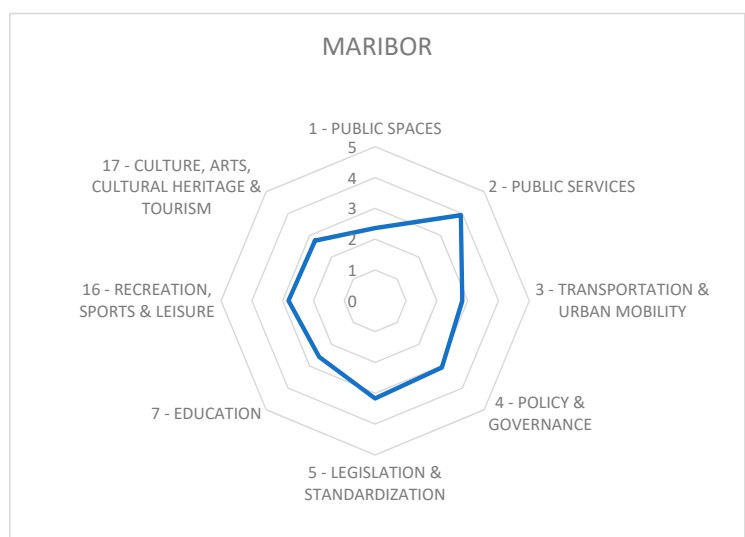

**Figure 1.** DIPI score for Maribor case study (selected areas).

In both cities, the testing was centrally managed, following the same protocol. Firstly, it was approved by the Mayor's office and a responsible department was assigned for the data collection, who then (in close collaboration with the researchers) took the leading role in coordinating data collection across city departments and external public offices. In this way, the Council for disabled people of Maribor, and the General Secretary of Pułtusk, respectively, greatly contributed to feeding the DIETool with the required data and obtaining the respective performance index DIPI.

The DIETool spreadsheet (in some cases, an area specific exert of the spreadsheet) had been sent for them to get acquainted with the nature of the required data. Usually, at least one on-site and/or one on-line meeting was organized. It took several interventions to guide the collaborators through the indicators and to obtain the respective data in a desired form. Although the initial plan was to test only about 5 areas per city, the research developed beyond our expectations. Both the cities demonstrated a high level of dedication and attempted to complete as many areas and indicators as possible. After 4 full months of data collection (June–September 2019), the team decided to complete the experiment in order to draw some preliminary conclusions. This pilot study brought interesting insights into the strengths and weaknesses, as well as challenges and opportunities to be addressed in the future regarding reliability and validity of the tool, the implementation process and the data collected.

### 3.3.1. Maribor Case Study

Maribor is the 2nd largest city of Slovenia, situated in the northeast of the country. In 2019, it had a population of 95,767 as a city settlement (not municipality), of which 47,144 are males and 48,623 are females. The number of persons with disabilities is not known. We estimated the number based on the results of the Eurostat household questionnaire (every 7th person or around 14.29% of population has some level of activity limitation [78], which accounts 13,685 of people with disabilities).

The Municipality of Maribor includes 10 departments and 27 organizational units. It has a city council with 45 councillors, of which 1 is with disabilities. The Council for disabled people was constituted in 2009 and represents the Mayor's consultancy body, as well as the network of all the

disability associations acting on the local level. It represents an important mediator and advocacy body for the rights of persons with disabilities in the city.

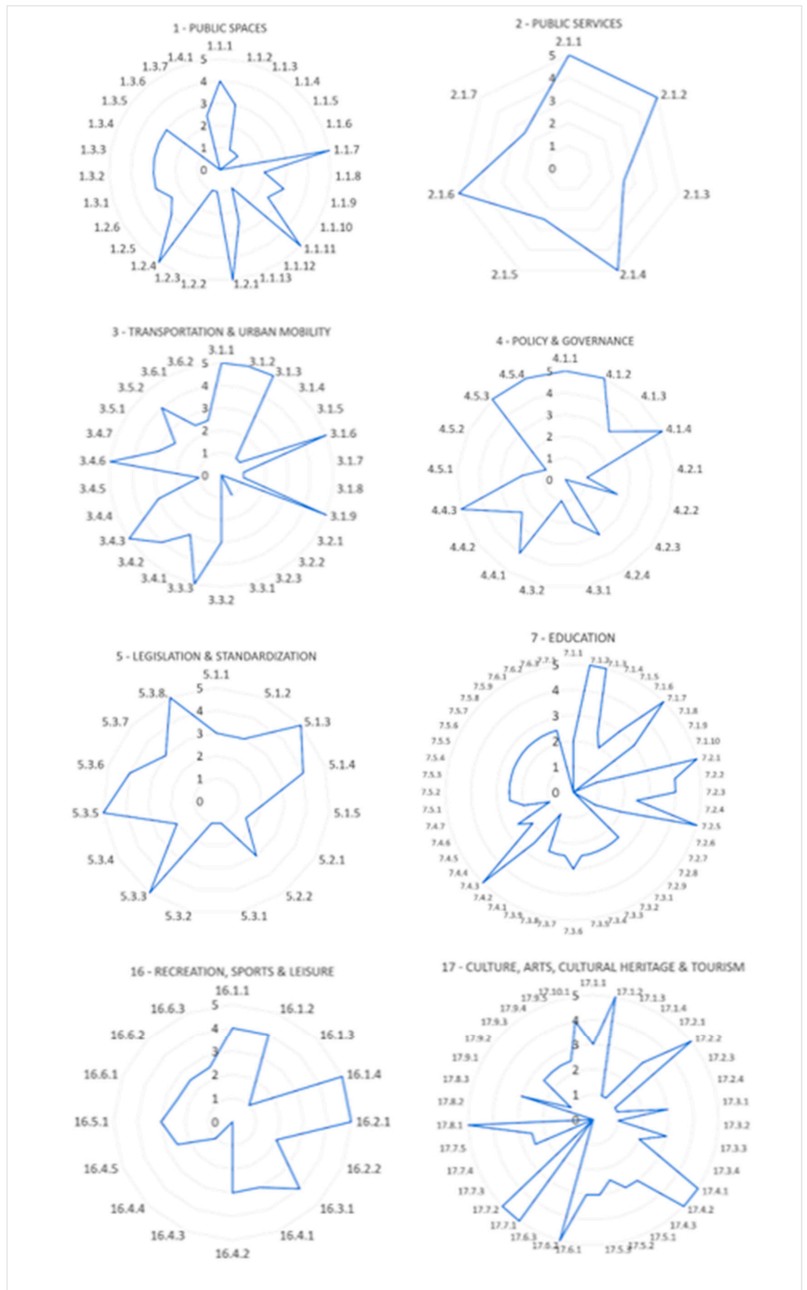

**Figure 2.** DIPI scores for Maribor for individual areas 1, 2, 3, 4, 5, 7, 16 and 17.

In the DIETool testing, the Council for disabled people of Maribor took the role of the internal coordinator. In total, 9 internal offices (Council for Disabled People, Mayor's Cabinet, City Council's office, HR office, Department for Communal Services, Transportation and Public Spaces, Department for Culture and Youth, Department for Sports, Department for Economic Activities, Department for Education, Schooling, Health, Social Services and Research Activities), and 13 external institutions were involved (local office of the Employment Service of the Republic of Slovenia, Regional Housing Fund, Social Work Center Maribor, Tourism Board Pohorje-Maribor, University of Maribor, University of Maribor Library, Maribor Public Library, Maribor Adult Education Center, local Police Administration,

local Chamber of Commerce, regional Association for Deaf and Hard of Hearing, local office of the Association for Students with Disabilities and Zavod PIP–Legal and Information Center Maribor).

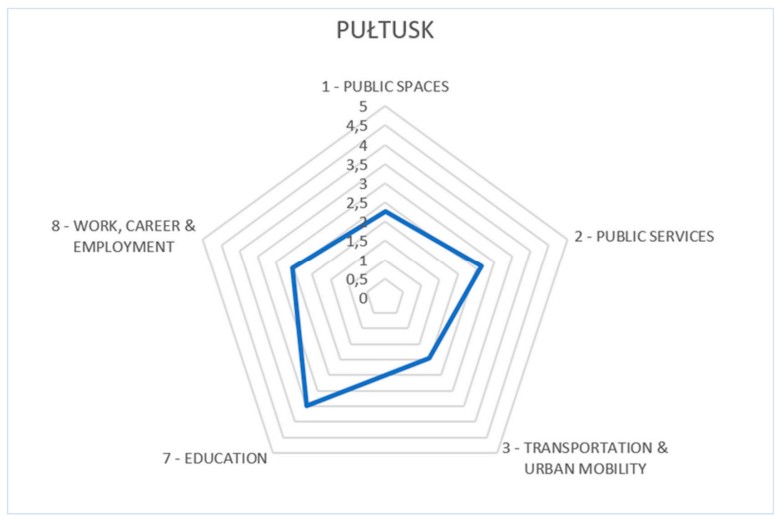

**Figure 3.** DIPI score for Pułtusk case study (selected areas).

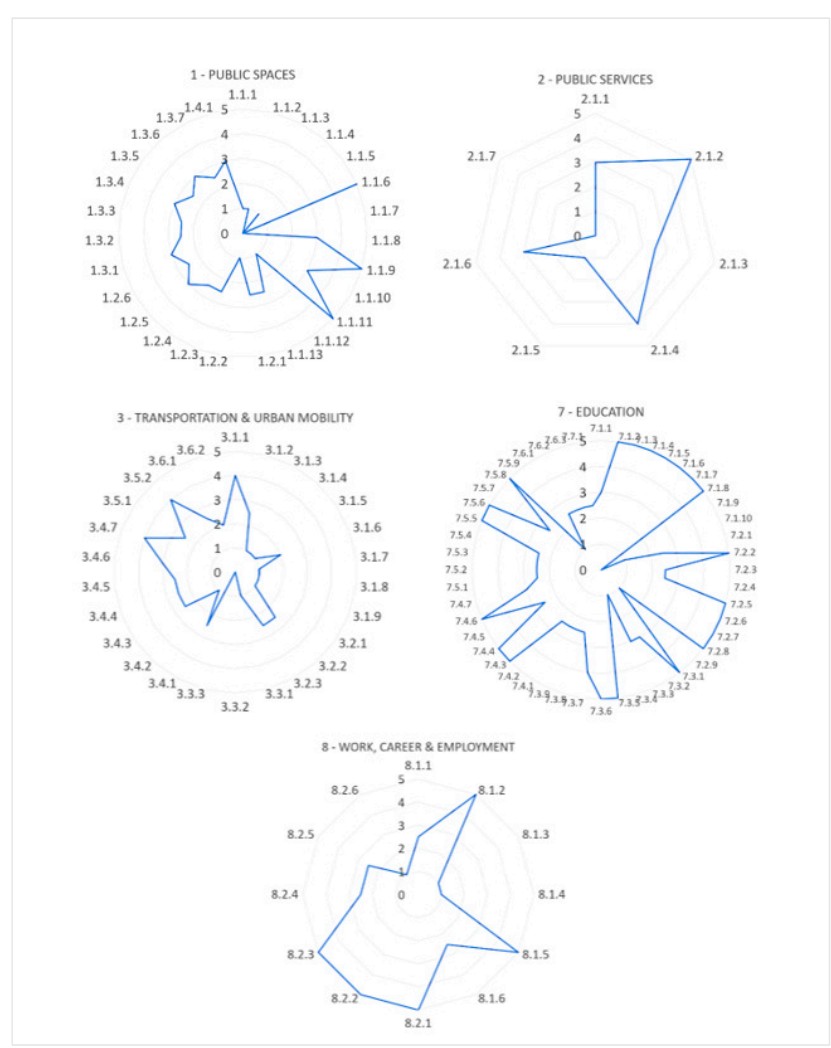

**Figure 4.** DIPI scores for Pułtusk for individual areas 1, 2, 3, 7 and 8.

3.3.2. Pułtusk Case Study

Pułtusk is a town in Poland lying by the river Narew, 70 kilometers north of Warsaw. It is the capital and the only town in the Pułtusk Poviat (district), in the northeast of the Masovian Voivodship. In 2018, it had a population of 19,431 in 2018, of which 9,298 are males and 10,133 are females. The number of persons with disabilities is not known exactly. However, based on statistics available for the Pułtusk District, we estimated the number of people with disabilities to 2,466 which amounts to 12.60% of the total population of the town. The structure of the Town hall consists of 7 departments, 6 offices, 5 independent workplaces and 10 organizational units. There is no organizational unit in the town dedicated specifically for people with disabilities. According to the Polish law, ensuring disability rights is a public task of district administrations, not towns. Only the towns with more than 50,000 inhabitants have a status of a "city county". As such, city counties cover the tasks on the commune and district (county) level. In the DIETool testing, the General Secretary took the role of the internal coordinator. In total, 6 internal departments/offices were involved: Department of Education and Promotion; Department of Organization and Supervision; Department of Civil Affairs, Department of Investment, Agriculture and Non-Budget Funds; Department of Land Management and Architecture; and Social Welfare Municipal Center. Moreover, 3 external institutions were involved in testing: District Labor Office, District Department of Education, and District Center of Family Support.

## 4. Results

This section addresses actual implementation of the tool in the two European cities, Maribor (Slovenia) and Pułtusk (Poland). It reflects some practicalities as well as challenges and benefits the studied cities may gain by tool implementation. Firstly, we elucidate the case studies in order to understand the context of the testing. Secondly, we represent the results obtained by each city. Finally, the relevant selected results are introduced and commented.

*4.1. Maribor Case Study*

4.1.1. Maribor Case Study Results

Maribor took a very dedicated holistic approach trying to provide information and gain insights into the performance in all the 20 areas. For this paper, we have chosen the 8 most representative and demonstrative areas (no. 1, 2, 3, 4, 5, 7, 16 and 17), where data has been provided for at least 50% of the indicators. The reasons why none of the areas reached 100% of completion are complex. Firstly, the city does not have a joint statistical office, a well-defined monitoring system or a profound e-governance approach. The data needed to be collected across a large array of resources and a wide network of stakeholders. Due to the limited staff capacity as well as limited amount of time, some of the indicators were therefore answered with "no data". The lack of city competences for obtaining the data (e.g., about the private sector) also contributed to such results. Nevertheless, enough data has been collected to be able to demonstrate the usefulness of the DIETool in the city of Maribor.

The Figure 1 shows the city's performance in the following 8 selected areas: 1-Public spaces, 2-Public services, 3-Transportation and urban mobility, 4-Policy and governance, 5-Legislation and standardization, 7-Education, 16-Recreation, sports and leisure, 17-Culture, arts, cultural heritage and tourism (a detailed list of indicators for each of the areas is available in Tables S1–S6, Table S8 and Table S9). According to Figure 1, Maribor performed average or above average in all the 8 areas with slightly better performance in the area 2-Public services. There are no areas with really poor performance.

However, the DIETool has been designed to explore deeper as of how the city performs in individual areas and individual indicators. The Figure 2 therefore, shows performance visualizations for each of the 8 areas (including all the indicators), where strengths and weaknesses can be identified in more details. They provide a quick diagnosis for Maribor and are to be further used to complement the DIETool spreadsheet for a thorough evaluation of each indicator that will indicate, where the

city needs to activate, as they wish, to improve disability inclusion performance and the DIPI index. For instance, when we look into the area 1-Public spaces (Figure 2–Inset 1 "Public spaces"; Table S1), we can see that Maribor performs very well (score 5) in indicators 1.1.7., 1.1.11., 1.2.1., 1.2.4. It performs well (score 4) in indicator 1.1.1., around average (score 3) in indicators 1.1.2., 1.1.9., 1.2.5., 1.3.1.–1.3.5. and poorly or very poorly (scores 0–2) in indicators 1.1.3.-1.1.6., 1.1.8., 1.1.12, 1.2.2.-1.2.3., 1.3.6.–1.3.7. Using the Excel spreadsheet for this specific area (Table S1) and using responses provided by the city officials, we can further analyze this result as follows.

In Maribor, the *Proportion (%) of wheelchair accessible parking payment machines (1.1.7.)* is 100% (scored 5), considering not only physical accessibility, but also the fact that public parking for persons with disabilities in the city is free of charge and alternative payment methods are provided (on-line and text message payment). The *Proportion (%) of accessible green spaces and* parks (1.1.11.) is also 100% and as such scored as very high (5), explaining that wheelchair accessibility is ensured to and within green areas, as well as a ground tactile guiding system is provided to the main city park, where some benches are also equipped with braille. Further on, very high score (5) has been assigned to the city in indicator 1.2.1. since the city provides a *participatory mobile app*, through which citizens (including persons with disabilities) can report accessibility issues in public spaces. The same score goes to indicator 1.2.4. because the city provides *tactile maps for visually impaired people*. Finally, the city reports that the *Proportion (%) of wheelchair accessible crossroads* (1.1.1.) is about 80%, which is considered as good (score 4).

Similar interpretation can be given for indicators given a score of 3, but with more consideration to the fact that these indicators need certain improvements in the future. For instance, *Proportion (%) of sidewalks accessible to wheelchair users* (1.1.2.) is only around 40%, which will undoubtedly need more dedication due to the lack of ramps, presence of barriers or damaged surfaces. *Proportion (%) of public toilet facilities accessible for persons with disabilities* (1.1.9.) still seems to be low and additional toilets will need to be planned, too. Only 42% or 3 out of 7 public toilets are accessible. These may be some of the reasons why the *Presence of people with disabilities in public spaces* (1.2.5.) may be rather around average (score 3). The reason may also lie in the average performance of indicators related to *Spatial planning* (1.3.1.), *Accessibility planning, management and evaluation* (1.3.2.), *Accessibility expertise among designers and architects* (1.3.3.), *Inclusion of accessibility requirements into public procurements* (1.3.4.) and/or *Inclusion of accessibility requirements into construction projects* (1.3.5.). For such performance, the city will need to rethink their planning, management and evaluation policy in the area of public spaces, spatial and environmental planning, and work towards a more elaborate, integrated and consistent system allowing for inclusion and accessibility principles to be incorporated throughout all the structures, services and processes.

The most intriguing part of the evaluation for the city comes where the performance is recognized as low or very low (0–2). An important set of questions appears, such as why the city is performing so low, which elements of accessibility and inclusion are missing and how they should be improved. Such examples are indicators 1.1.3.–1.1.5., which all refer to *accessibility of public spaces for visually impaired people*. 1.1.3. and 1.1.4. demonstrate that *crossroads* and *sidewalks are equipped with a tactile guiding system* in only about 10%, whereas 1.1.5. shows that only 9% of *traffic light crossroads are equipped with audio signals for visually impaired people.* These indicators clearly show a joint nominator, pointing out that the city might have a poor policy for ensuring accessibility in public spaces in terms of the needs of visually impaired people. A few other indicators with low or very low performance may be mentioned. The indicator 1.1.6. shows a low *number of wheelchair accessible parking spaces*, which speaks about the initial number of designed dedicated parking spaces as well as their actual accessibility (without barriers, with manoeuvring space, access to a sidewalk, etc.). The indicator 1.1.12. points out that the city might not have enough *wheelchair accessible playgrounds for children* and may need to consider not only arranging an access to the playing surfaces, but also providing wheelchair-dedicated swings and other accessible facilities, etc. The indicator 1.3.6. refers to *accessibility requirements for events organized in public spaces*, which may or may not be included in the general conditions while obtaining

permission. In Maribor, unfortunately, there are no such requirements and actual accessibility provision is a result of how knowledgeable, aware and willing organizers of the events are. The DIETool clearly shows absence of disability inclusion policy in the area of organization of events, and, as such, aims at encouraging the city towards future measures.

### 4.1.2. Maribor Case Study Observations

Maribor, as a city, is one of the smaller among middle-sized cities in Europe. It has low investment rates and relatively low innovation in the public sphere. Open and digital governance is still under construction without any real implementation yet. These factors may clearly outbalance the generally solid DIPI performance. As seen from the graphs, numerous indicators have been assigned values above average (3–5) but have been outbalanced by others below average (0–2).

Nevertheless, with all its efforts, the city of Maribor demonstrates a firm tradition of disability inclusive city management. Firstly, they were immediately willing to participate, provide the data and explain specific indicators. Secondly, they have had the Council for Disabled people as a part of the city's organizational structure for around 10 years. Thirdly, they have tried to complete the whole DIETool and were thus in contact with the researcher on a weekly basis to find ways of providing the data. They accompanied the DIETool Excel spreadsheet with a word document, providing descriptive contextual information area by area, indicator by indicator, which will give them a greater insight into the current state and the needed improvements.

Some challenges in obtaining the data may be worth mentioning. In Maribor, it was difficult to get even an overall idea of the situation referring to the private sector (private employers, private educational institutions, private cultural, sports and tourism services, banks and financial services etc.), unless the respective city department had been involved with private stakeholders already in the past. Additional challenges appeared in some cases, when the working area is regulated on the national level (e.g., for secondary education) or is under the power of another public institution (e.g., University of Maribor for higher education). In such cases, in Maribor, they tried to find information through publicly available resources (e.g., websites, reports, articles) or they contacted external stakeholders (e.g., University of Maribor, Employment Service). These efforts showed that the data collection and disability inclusion monitoring is greatly dependent on many factors, including the level of collaboration and partnerships across different sectors and stakeholders, creating a cohesive society.

### 4.2. Pułtusk Case Study

### 4.2.1. Pułtusk Case Study Results

Pułtusk focused on the areas 1-8, but in the later stages tried to find more data also on further areas. For this paper, we have chosen the 5 most representative and demonstrative areas (no. 1, 2, 3, 7 and 8), where data has been provided for at least 50% of the indicators. In none of the areas, 100% of indicators were unfortunately collected. One of the reasons is the fact that disability inclusion by Polish law is coordinated by the city counties. As such, disability inclusion is thus mostly regulated on the regional level. Consequently, Pułtusk does not have any disability inclusion related office or a disability inclusion monitoring system. The lack of city competences in some of the areas (e.g., that are regulated on the regional level or are a matter of a private sector) greatly contributed to the lack of data too. Nevertheless, the DIETool can still demonstrate the city's performance in the selected areas. The Figure 3 shows how well Pułtusk is performing in the following 5 selected areas: 1-Public spaces, 2-Public services, 3-Transportation and urban mobility, 7-Education and 8-Work, career and employment (for detailed list of indicators for each of the areas see Supplementary Materials, Tables S1–S3, Table S6 and Table S7). According to the Figure 3, Pułtusk performed above average in the area 7-Education (app. score 3,5) and around average in areas 1-Public spaces, 2-Public services, 8-Work, career and employment (app. score 2,5). There is one area with poor performance, that is 3-Transportation and urban mobility (app. score 2).

Similarly to the previous example of the Maribor case study, we provide 5 area specific insets (Figure 4) also for Pułtusk. They give an opportunity to explore the results in more details and point out indicators' specific strengths and weaknesses. When combined with the DIETool spreadsheet, the initial diagnoses can thus be further analyzed through each indicator.

For instance, when we look into the area 8-Work, career and employment (Figure 4–Inset "8-Work, Career & Employment"; Table S7), we can notice that Pułtusk performs either very well (score 5) in indicators 8.1.2., 8.1.5., 8.2.1., 8.2.2., 8.2.3. or it performs very poorly (scores 1) in indicators 8.1.3., 8.1.4., 8.2.6. It is worth noting that such bipolarism also occurs in other areas in which the city achieves very good results for some indicators and weak for others.

Using the Excel spreadsheet for this specific area (Table S7) and using responses provided by the city officials, we can further analyze this result as follows. In Pułtusk, *Unemployment rate (%) of individuals with disabilities against unemployed individuals without disabilities* (8.1.2.) is 3.38%. It means that in the city, there are 42 registered unemployed persons with disabilities out of 1.242 registered unemployed persons without disability. Besides that, the *Proportion (%) of youth with disabilities (aged 15–24 years)* not in employment nor in education and training against all young persons with disabilities (8.1.5.) is 4.76 %. It means that only 4 youth with disabilities out of 42 of totally registered persons with disabilities are not in employment nor in education or training. The values obtained for the above indicators reflect that the professional integration of people with disabilities, carried out by the Poviat Labor Office, is quite effective. The second group of indicators in which Pułtusk also performs very well (score 5) are indicators: *Availability of programs for disability employment development* (8.2.1.), *Availability of funding for employers (for employment, for ergonomic solutions, rehabilitation and training, etc.)* (8.2.2.) and *Availability of funding opportunities for disabled people (for self-employment, individual training, etc.)* (8.2.3.). Here we see the interconnection between the previous two indicators (8.1.2. and 8.1.5.) with the last three (8.2.1.–8.2.3.). It is probably the availability of active programmes and funds for people with disabilities that contributes to such a good result and makes the number of registered persons with disabilities relatively low.

However, one must bear in mind, when addressing the whole picture, that there was a lack of data for some indicators, such as: *Proportion (%) of unemployed individuals with disabilities against all active persons with disabilities* (8.1.1); and *Proportion (%) of individuals with disabilities living below the national poverty line* (8.1.6.). The three indicators, for which the city performs poorly (score 1) are: *Proportion (%) of individuals with disabilities in long-term unemployment against the unemployed persons with disabilities* (8.1.3.); *Proportion (%) of temporarily employed individuals with disabilities against the total no. of employed persons with disabilities* (8.1.4.); and *Level of satisfaction among employees with disabilities (working climate, equity, available ergonomic solutions, staff training, support provision, etc.)* (8.2.6.). In the city, there are 27 persons with disabilities in long-term unemployment among 42 registered unemployed persons with disabilities (64%) and 36% temporarily employed. These results confirm that the issue of professional activation of the long-term unemployed, including people with disabilities is very complex. Among the reasons for this situation, we can find the lack or inappropriate qualifications, the lack of jobs in the place of residence as well as low mobility, possibly low personal motivation, and/or family problems. In many cases, these causes an overlap. The city may need to explore further how to overcome these numbers. Finally, the level of satisfaction among disabled employees is around 20%.

A further questionnaire could tell whether the non-satisfaction is a result of insufficient ergonomic solutions, attitudinal barriers, economic reasons, non-suitability of work or the lack of opportunities for career advancements. Summarizing the results for area 8, we can say that DIETool clearly indicates difficulties in monitoring the employment of people with disabilities. Although in the case of Pułtusk town, conducting labor market policy is the task of the poviat, this tool shows in what areas the city could undertake improvement actions related to work and employment of the persons with disabilities, in particular, those long-term unemployed and those already employed.

### 4.2.2. Pułtusk Case Study Observations

The DIETool testing in the city of Pułtusk has highlighted several challenges and limitations in collecting the data, too. The first and most important issue turned out to be that small cities in Poland, such as Pułtusk, constitute parts of poviats (county), and only these have the legally assigned competences regarding social inclusion, in particular, related to persons with disabilities. Additionally, other related competences such as health protection and hospital management, organization of upper secondary schools, or running labor market institutions belong to the poviat administration and not to the city administration. Therefore, using a tool such as DIETool, in the case of small cities, requires the inclusion of the poviat administration in the research process. The second most important limitation associated with the first one is the fact that statistical data on social inclusion are collected only at the poviat level. The general difficulty is, therefore, to disaggregate the data in such a way to obtain data for the city. On the territory of the Pułtusk Poviat, which includes six communes, there is only one city, Pułtusk; hence the starting point for many calculations as part of the research was the percentage share of the city's population among the poviat residents (app. 37 %). Thirdly, within the research, we detected a lack of data collected by the city on the DIETool indicators related to private and non-governmental sectors.

These were among other reasons why Pułtusk, as a small city without a department dedicated specifically to the area of disability inclusion, faced difficulties completing the DIETool within the timeframe of the pilot study.

## 5. Discussion and Conclusions

Recent studies [2,9,12,14,19] show that contemporary city concepts, such as the smart and sustainable city, are not fully ready to meet the requirements for an inclusive city for all. That is, a city inclusive of all their inhabitants, including those with disabilities and diverse impairments [1,2,30]. Within current conceptualizations, disability inclusion gets lost in overall inclusion dimensions, perceived either as inclusion for all, economic, social or racial inclusion [14,15]. Social equity and social inclusion dimensions within social sustainability debates [36–43], although are important paradigms, can neither respond to the complexity, diversity and specific nature of disability inclusion. This creates a need for further discussions as of what an inclusive city truly means, and specifically, how to address the inclusiveness of vulnerable groups of citizens, such as those with disabilities [2,17,29]. Modern urban [60–65] and disability rights agendas [3–5] with respective political, legislative and standardization frameworks provide a starting point for understanding what a disability inclusive city is. However, a lack of disability specific evaluation, monitoring and implementation tools may be one of the reasons why cities are still unable to follow these ambitious agendas and legislative frameworks to bring them to practice.

The DIETool presented herein attempts to translate political and legislative requirements into main inclusion and accessibility principles, and holistically integrates them into a single tool covering diverse a) disability types; b) city settings (areas); and c) services, products, environments and information. It responds to the absence of disability inclusion specific evaluation and monitoring systems within current smart, sustainable and inclusive city initiatives [21,24,36,37,44–55], and creates a vision of a disability inclusive city.

The results of the testing in Maribor and Pułtusk confirmed the lack of such tools in practice. Most of all, they proved that the evaluation process for cities is demanding, not only in terms of time and staff capacity, but also in terms of understanding disability inclusion and its complexity. Other issues are both the level and the scope of city competences regarding disability inclusion, as well as data collection possibilities. A good coordination across internal city departments, as well as partnerships with external stakeholders are needed, and still the data is difficult to obtain. Nevertheless, we noticed that a designated disability inclusion office may bring more success in getting informed about the actual level of inclusivity. Such was the case of Maribor, where the presence of the Council for Disabled People showed how a small organizational change and a tradition in ensuring disability rights within

the city organizational structures can make an important difference. In Maribor, the dedication to complete the DIETool was, therefore, very high, and the experimentation resulted in 8 areas completed with more than 50%, and an additional 8 areas with at least 25% completion rate.

For Pułtusk, on the other hand, firstly the collection of data was poorer. Only 5 areas were completed with at least 50% of indicators, and an additional 2 areas with at least 25%. This is hardly surprising, having in mind that it is a small city of only app. 19,500 inhabitants without any Disability Rights office. As such, it has limited staff capacity and disability rights related competences, thus the data collection was even more demanding.

Despite this brief comparison, the aim of the DIETool and DIPI, at this point, is not to make cities competitors but rather to help an individual city to see where their strong and weak points are. In fact, as we see in the case of Maribor and Pułtusk, the two cities are non-comparable, although they are both European cities under the powers of the same EU directives and frameworks. Nevertheless, they have their national and local specifics. They are of different size, with different economic development and different disability rights traditions, as well as different organizational, legislative and socio-cultural backgrounds to name only a few. Diverse internal and external factors may be affecting the city governance, which, in turn, affects the situation of people with disabilities in the city and impacts the DIPI score.

In both cities, however, we faced challenges with obtaining the data. The reasons were multifaceted. Firstly, one of the reasons may be the nature of the indicators, although our aim was to simplify the extensive set of legal accessibility and disability rights requirements and standards, not to make them more complicated. Secondly, absence of a statistical office and a monitoring system in the city may reflect the lack of detailed analytics needed to feed the DIETool. Thirdly, the city may have divided competences across specific working areas, thus may not be able to collect the data. For instance, many disability inclusion related aspects in Poland are managed on a regional level by counties (poviats), and in Slovenia, some areas are centrally managed by the national government. In addition, a wide network of stakeholders may be needed to collect the data from the private sectors. On the other hand, both the tested cities were recognized to have a very low level of digital governance. A digital upgrade may thus represent an interesting and needed vision for the future of the DIETool to help cities automize parts of the data collection, analysis and management processes, while at the same time disaggregating data for monitoring disability inclusion. Furthermore, data reliability and validity may need to be discussed further, too. The pilot tests could not give answers to this challenging field.

The two pilot case studies, however, should not be considered as a proof of concept, but rather as the DIETool demonstration as of how to evaluate and monitor disability inclusion performance and stimulate cities to become more sensitive and holistically conscious of disability inclusion. We can consider it, as a proof, that such a demanding evaluation process can be done. Despite the challenges in data collection in Maribor and Pułtusk, we cannot claim that the data cannot be obtained more extensively in other cities with different organizational structures, management systems, different disability inclusion policies and advanced digital governance. For instance, it would be interesting to explore the DIETool in a context of a large, highly digitized and citizen inclusive smart city [46], such as Singapore (Singapore), Zurich (Switzerland), Copenhagen (Denmark), Vancouver (Canada), Amsterdam (Netherlands) or Vienna (Austria).

The DIETool, overall, demonstrates several strengths, benefits and opportunities. Firstly, it sums up and simplifies a complex set of disability inclusion rights and accessibility requirements across diverse areas of city life. As such, it supports the process of learning, raising awareness and gaining a profound understanding of what disability inclusion really means.

Secondly, as a disability specific evaluation and monitoring tool, it attempts to fill in the gap within the existing city metrics, which currently cover attributes, such as smartness, (social) sustainability, quality of life, creativity, etc., but not yet disability inclusion. When applied to practice, the DIETool points out to the strengths and weaknesses of a city's performance in a sensitive area of disability

inclusion and by doing so, it guides the city towards designing informed corrective measures for an accessible and disability friendly city.

Thirdly, the DIETool acts as a toolkit and guidelines. It not only sets the requirements but also envisions a set of alternative techniques (e.g., questionnaires, focus groups). These aim at a) supporting the demanding data collection processes, and b) obtaining the viewpoint of other stakeholders (citizens, non-governmental and private sector). This may increase reliability and validity of the tool in obtaining the data, as well as may help objectivize, validate and, as such, balance out the results that may be biased just because they were collected only from city administrations. By doing so, the tool provides a mechanism to control validity and reliability risks in terms of possible conflicts of interest and subjectivity. Unfortunately, our preliminary pilot study did not incorporate such validation tests with direct users yet, as it was conducted with two main aims: a) to see whether such a tool could be accepted by city administrative decision, and b) to see whether the data for the envisioned indicators could be obtained, to what extent and in what form, in order to further approach its reliability and validity issues. On the one hand, we proved that the tool can be accepted even though it demands commitment, effort and time. On the other hand, the data can be collected to a certain degree to feed the tool, even though many challenges have been faced.

Fourthly, the tool requires a proactive approach. The joint efforts across different city departments, institutions and diverse range of stakeholders to obtain the data will undoubtedly create stronger partnerships also for future actions, improve data collection processes and, in turn, improve data reliability and validity, too.

Fifthly, it may also help setting inclusiveness as a norm and one of the main principles within the work of all the city's departments. This could bring an important opportunity for reaching a long-term commitment and socially sustainable approach towards inclusive city design. As shown in the case study of Maribor, the DIETool can be used to conduct a preliminary (or periodical) analysis and can feed annual reports on the equal opportunities for persons with disabilities. It can support the design of a local disability inclusion strategy and provide the basis for an action plan towards a disability inclusion in the city. Nonetheless, the DIETool can be used both as a complete tool across all the 20 areas and as a tool to detect weaknesses in specific working areas, aiming at guiding specific city departments in improving their disability inclusion performance.

Finally, the DIETool demonstrates strengths, also speaking in terms of software quality design. For it to be accepted by the cities, we designed it in a way that it fulfils certain quality properties following the product quality model proposed by the standard ISO/IEC 25010:2011 The DIETool, therefore, fulfils quality in terms of 8 characteristics: Functional suitability, Reliability, Performance efficiency, Usability, Security, Compatibility, Maintainability and Portability [79] (p. 10). During the pilot testing, the tool achieved a good level of acceptance among city staff. It was conceptualized in a way that assumes a basic knowledge of disability inclusion, but it does not require advanced knowledge for its comprehension. In fact, the tool itself is a source of knowledge for the user and, as a result, provides learnability and promotion of disability inclusion. Using the tool should, thus, not be a barrier. Although the process of data collection for each of the indicators may be challenging, very time consuming and requires extensive work, once the data is collected, the tool enables a quick entry, immediate DIPI score calculation and the provision of corresponding graphical visualizations (both for the overall DIPI score, as well as area specific DIPI scores). This enables an immediate diagnosis and efficient analysis of results.

The DIETool itself guarantees high reliability for the city staff due to high probability to maintain its level of performance and high performance of failure-free operation in the environment of the public administration. DIETool, due to its design, can also continue operating properly in the event of inability to use one of its independent components (areas). In fact, it has been designed so to be able to follow the overall performance across all the areas, across a few selected areas and/or only within individual areas. However, over time it may be necessary to update the tool, e.g., by incorporating mechanisms

for validity risk control, developing a weighting system and ensuring subsequent up-to-date versions, all that taking into account the complexity and diversity within the fast-paced changing city life.

Summarizing the pilot case studies, although not conducted as classical big-scale verification tests, have helped us point out some of the limitations, drawbacks and challenges to be addressed in the future. These may refer to: (a) the implementation process (e.g., the need for coordination, commitment, effort, time, partnerships, risks of subjectivity, conflicts of interests, validation process, involving direct users, etc.), (b) the data collection process (e.g., data collection methods (primary/secondary data), data validity issues (lack of data, dispersed, incorrect, subjective, non-comparable, non-consistent data, for instance, in the scope; geographical coverage, time span, status-related etc.), and (c) the DIETool with its methodology (e.g., the need for a weighting system, mechanisms for reliability and validity risks control).

For instance, the weighting methodology will need to be explored considering various internal and external factors. Prioritization of indicators and areas, as well as compensation considering diverse distorting factors may need to represent one of the first steps towards improving the reliability and validity of the tool, its methodology and its results. One of the current limitations of the tool is its non-comparability across diverse city contexts. Thus, city-specific factors may need to be studied. These factors, among others, include the country of origin, the city's development stage, its size, overall maturity level, etc. For instance, the scoring scales for crime rates, employment rates etc., may need to be different for differently developed countries (non-comparability across countries). Optimal targeted performance for cities of different size may be different, too. We cannot compare a small town of Pułtusk and a middle-sized city of Maribor with a metropolitan city of a complex structure inhabited by several million people on the one hand, and with a greater financial and organizational capacity on the other hand. In the future, we may think of a tool with pre-set country specific and/or size specific profiles. Such profiles would include different scoring reference rates and possibly even different indicators, depending on the national and size specific requirements.

Pilot testing in the two cities did not bring answers in many aspects listed above. It is a new tool, process-related and therefore "a-live". Thus, the challenges will need to be addressed gradually. Over time, however, the number of inconsistencies will decrease as it matures. In the future, we may even envision it as an integral component of a complex smart city platform with interconnected subsystems/areas (e.g., 20 as in the DIETool) providing and exchanging information in real time on demand. Each of the areas would be fed with content relevant for the citizens, enabling them to get the right information at the right time, get engaged with city authorities, collaborate in city surveys or answer questions at points of interest. On the other hand, it would enable cities to have a direct and channelled citizens' feedback supported with an automated data collection and analytics system. The DIETool could serve as a monitoring tool within the platform, benefitting all the city departments with automatically fed information, as well as enabling the creation of additional surveys, reports and finally the DIPI performance score.

With the presented work, we hope to have provided a practical tool that will advance research and practice on how to design a socially sustainable city concerning questions of inclusiveness for all, and by that, stimulate current smart and sustainable city initiatives to consider disability inclusion as an important element of any contemporary city. Our vision is to bring accessibility and inclusion into everyday lives of people with disabilities, for which, each city subsystem, service, product, process or information needs to be designed, evaluated and monitored in line with best disability inclusion practices.

**Supplementary Materials:** Supplementary materials consist of Tables S1–S9. They are available online at http://www.mdpi.com/2071-1050/12/4/1378/s1.

**Author Contributions:** N.R., M.S., A.B. and B.G.M. together conceived, designed and wrote this paper. The DIETool was designed by N.R. and M.S. and evaluated by A.B. and B.G.M. The testing was then implemented in the two cities by N.R. and M.S. The collected data was analyzed and validated by all the authors. Visualizations were prepared by A.B. and N.R. The project was managed and administered by N.R. All authors have read and agreed to the published version of the manuscript.

**Funding:** This research has been supported by the European Union under the H2020 Marie Skłodowska Curie Action (ref. N° 665959), and by the Slovenian Research Agency (research core funding N° P5-0100).

**Acknowledgments:** The authors would like to thank the City of Maribor (Slovenia), especially the Council for Disabled People of Maribor, and the City of Pułtusk (Poland) for their dedicated approach and full engagement in the study.

**Conflicts of Interest:** The authors declare no conflict of interest. The funders had no role in the design of the study; in the collection, analyses, or interpretation of data; in the writing of the manuscript, or in the decision to publish the results.

## Abbreviations

| | |
|---|---|
| ANED | Academic Network of European Disability Experts |
| DIETool | Disability Inclusion Evaluation Tool |
| DIPI | Disability Inclusion Performance Index |
| DOTCOM | Disability Online Tool of the Commission |
| EC | European Commission |
| EDF | European Disability Forum |
| EU | European Union |
| ICT | Information and Communication Technologies |
| PWD | person with disabilities |
| SDGs | Sustainable Development Goals |
| UD | Universal Design |
| UNCRPD | United Nations Conventions on the Rights of Persons with Disabilities |

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
