# Peer review of "Measuring Disability Inclusion Performance in Cities Using Disability Inclusion Evaluation Tool (DIETool)"

_sustainability, doi:10.3390/su12041378_

Round 1

Reviewer 1 Report

This is technical paper presenting a classification tool measuring cities capability to cope with disabled. The paper presents two case cities from Europe. The literature review is limited and biased for these making generalizations regarding the essence of a ‘European city’. The paper lacks critical discussion of the validity and reliability of the constructed tool. The short remark at the end of conclusions does not do the job well enough but mentions the obvious point that the tool ignores context of specific cities that are significant determinants of how cities are planned and structured. On the other hand, I think that the amount of work has been extensive in terms of data collection.

Author Response

RESPONSE TO REVIEWER 1:

The reviewer expresses the need for bringing the literature review beyond generalizations regarding the essence of a ‘European city’ and emphasizes the importance of discussing critically the validity and reliability of the tool. We sincerely thank the reviewer for these valuable comments that might further improve not only the paper but also the future development of the tool. We attempt to respond to the proposed suggestions to the best of our abilities concerning the current degree of the research. The responses are as follows:

1. Literature: By adding a subsection 3. City-specific context we attempt to overcome the generalizations brought by previous global and European-related literature and give the reader a closer look into the city context of the two case study cities. Within this section, we gradually bring in a few important documents, such as national strategies and action plans for both countries, as well as local strategies, analysis and action plans that influence our specific cities’ contexts. Additionally, we mention interesting disability inclusion related initiatives that in the past influenced the ways disability inclusion is nowadays perceived and governed in the two cities Maribor and Pultusk. Please see subsection 2.3. for more details.

2. Reliability and validity of the tool have been discussed with more details in section Discussion and Conclusions. Please see the highlighted changes on pages 17-20.

In addition, we wish to emphasize that the paper should be recognized as a pilot study conducted with two main aims: a) to see whether such a tool could be accepted by city administrative decision making because it demands commitment, effort and time, and b) to see whether the data for the envisioned indicators could be obtained, to what extent it could be obtained and in what form in order to further approach its reliability and validity issues.

We were also interested in what kind of challenges the data collection would bring, which we had already discussed in the initial version of this manuscript. Please see page 17.

Furthermore, it needs to be taken into account that the tool itself was initially designed a) firstly, to simplify the vast array of requirements within political, legislative and standardization frameworks, b) secondly, to help cities see disability inclusion more holistically by bringing these dispersed requirements together, and c) thirdly, to encourage city governments towards taking introspection, developing sensitivity and taking action towards actual accessibility in practice. As such, the tool, the collected results and the outcoming score are currently valid and comparable only within one specific city context (e.g. Maribor or Pultusk), where the score serves as a measure to help attract cities by simplifying and visualizing the complex field of disability inclusion. Neither the DIETool nor the DIPI index were designed with the aim to become a purely quantifiable ranking score. We rather aim at conducting a qualitative small and soft data-oriented research as a means to promote the inclusion of individuals with disabilities, not to grade or rank cities. In this respect, the tool envisions qualitative validation tests with citizens that will help objectivize and validate the results through the eyes of direct users not only city administrations.

Having said that, the pilot study has helped us point out to some of the challenges to be addressed in the future within: a) the implementation process (e.g. the need for coordination, commitment, effort, time, partnerships, risks of subjectivity, conflicts of interests, validation process, involving direct users etc.), b) the data collection process (e.g. data collection methods (primary/secondary data), data validity issues (lack of data, dispersed, incorrect, subjective, non-comparable, non-consistent data, for instance in the scope, geographical coverage, time span, status-related etc.), and c.) the DIETool with its methodology (e.g. the need for a weighting system, considering mechanisms for reliability and validity risks control).

3. Data collection: We thank the reviewer for acknowledging the amount of work that has been done in feeding the DIETool within real city contexts. In fact, getting cities interested in participation, as well as the availability of the needed data were our main goals, as well as concerns during the development stage. Both have proven to be achievable, of course with a set of challenges discussed above and in the paper.

Reviewer 2 Report

I applaud the authors’ efforts to bring objectivity to administrative decision making. I teach a graduate course precisely on this topic.  I also applaud the authors’ efforts to develop a means with which to promote the inclusion of individuals who have disabilities. I recognize this as a pilot effort, and I believe that publication is warranted.   I offer a couple of points for consideration.

In a sense, you have created a scale.   If you have not yet collected information on the reliability of the scale, this would be a good next step.   Clearly, reliability will be the first question asked by potential users of the scale. If you have reliability data, please include it. If you don’t yet have reliability data, you might propose how you will collect these data. Weighting – I would suggest that you avoid this even though it has substantial face validity. Statistical research has suggested that weighting loses all impact after about 10 items in a scale.   As I understand it, the weights are essentially overpowered by the covariance matrix. I once ran a study of weighted and unweighted versions of a scale that had about 36 items.   The correlation between weighted and unweighted versions was .99.   This is apparently the common finding.   On the other hand, the addition of (even meaningless) weights might be viewed positively by potential purchasers.   Not useful mathematically, but maybe a good sales point. I might look for a way to reference the United Nations position statements on disability ahd inclusion.   Not essential.

Author Response

RESPONSE TO REVIEWER 2:

We sincerely thank the reviewer for acknowledging our work, specifically in the part, where lots of efforts have been done to bring the tool to practice. With our work beyond this paper we have been long aiming at encouraging city governments towards taking introspection, developing sensitivity and taking action towards holistic implementation of accessibility and disability inclusion principles into practice. In fact, getting cities interested in participation, as well as the availability of the needed data were our main goals, but also concerns during the development stage. Both have proven to be achievable in the two pilots, of course with a set of challenges to be addressed in the future.

We also appreciate the reviewer for sharing their experiences and expertise with us. It would be interesting to experience the exchange in person, possibly for potential future collaboration. Their valuable comments have helped us further improve not only the paper but also reconsider some of the challenges to be addressed during the future development of the tool. The full list of responses to the comments is provided here:

1. Reliability and validity-related discussions have been incorporated into Section 5: Discussion and Conclusions on several occasions (pages 17-20). There, the reviewer will find highlighted additionally outlined challenges that the pilot study pointed to, including the reliability of the tool, the scale and the data.

Unfortunately, during our pilot study no systematic reliability nor validity test was conducted. Therefore, we do not have reliability data yet. As explained above, we were mostly interested whether such a tool could in fact penetrate into city administrations and if the data could be collected, to what extent and in what form. The tool itself was initially designed a) firstly, to simplify the vast array of requirements within political, legislative and standardization frameworks, b) secondly, to help cities see disability inclusion more holistically by bringing these dispersed requirements together, and c) thirdly, to encourage city governments towards taking introspection, developing sensitivity and taking action towards disability inclusion in practice.

As such, the tool, the collected results and the outcoming score are currently valid and comparable only within one specific city context (e.g. Maribor or Pultusk), where the score serves as a measure to help attract cities by simplifying and visualizing the complex field of disability inclusion. Neither the DIETool nor the DIPI index were designed with the aim to become a purely quantifiable ranking score. We rather aim at conducting a qualitative small and soft data-oriented research as a means to promote the inclusion of individuals with disabilities, not to grade or rank cities.

In this respect, the tool envisions qualitative validation tests with citizens that will help objectivize and validate the results through the eyes of direct users (survey, interviews, focus groups) not only city administrations. This may in the future help avoid risks of subjectivity and conflicts of interest. In addition to reliability and validity challenges, reliability of the tool as a system is briefly explained in the Discussion section on page 19.

2. A weighting system has been considered as it gives validity to the tool and as the reviewer already emphasized ‘might be viewed positively by potential purchasers’. However, as explained above, neither the tool, nor the index were designed as a purely quantifiable score. Moreover, the aim of the tool is to provide cities a self-assessment tool and encourage them to explore deeper into each of the areas and each of the indicators in a qualitative way. Because our aim is not to create a ranking system, the weighting although may be considered as a potential added value, will be of lesser priority in DIETool upgrade.

3. The United Nations Disability Inclusion Strategy has been referenced. We suppose that the reviewer referred to this document, available here: https://www.un.org/en/content/disabilitystrategy/assets/documentation/UN_Disability_Inclusion_Strategy_english.pdf (February 2020).

Again, with great appreciation we thank the reviewer for applauding our work. It would be interesting to join efforts in any future occasions. With best wishes.

Reviewer 3 Report

The thematic of the manuscript is very relevant as regards social inclusion and inclusive city development. The topic presented in the manuscript is relevant for the journal, and the case studies presented could provide useful insights and advancements to feed the debate on such a topic. However, I recommend publication just after some revisions that are required especially in the Results section.

Specific comments

The first 2 sections provide enough background to readers. However, I have some comments for their improvement:

It should be better for readers to simplify the sentence in lines 69-70: “[…], such as how to design socially sustainable cities concerning questions of inclusiveness also in terms of the needs of those with diverse impairments”. Line 73: specify what are you testing in this section (I suppose the DIETool). Line 86: it should be better to write “get often excluded” instead of “get excluded”. In paragraph 2.1 the paradigms of social equity and inclusion are introduced (and also mentioned in other following sections of the manuscript); it should be useful for readers to briefly explain what these two paradigms are just after their introduction.

As regards Section 3 and Section 4, I have the following comments:

In Table 1: in the sentence “CHRACTERISTICS/REQUIREMENTS/EXAMPLES FOR INDICATORS' DESIGN*” there is a typo in the word “characteristics”. The information reported in the column “CHRACTERISTICS/REQUIREMENTS/EXAMPLES FOR INDICATORS' DESIGN*” of Table 1 should be better linked with the one reported in the tables of indicators (in Appendix), e.g. reporting in Table 1 the macro-categories of indicators of each area (such as 1.1 accessible infrastructure 1.2 participation and communication, etc. for Area 1 public spaces and built environment) and then providing some characteristics/requirements/examples for each one of these macro-categories. In paragraph 3.2 you should specify that the overall performance score of each Area is given by the average score (I suppose) of all the indicators calculated within that area. Line 330: closing bracket missing. In Figure 1: it seems that there is a missing score for “Area 1 Public spaces” since there the blue line forms a straight line from the score of “Area 17 Culture, ...” to the score of “Area 2 Public services”. Lines 362, 366, 375: error with reference sources. Line 369: it should be better to specify what do you mean with “explore deeper” (i.e. explore what?). Figure 2 and Figure 4: a lot of indicators’ numbers overlap, making it difficult to interpret the graphic for readers. Consider replacing these figures with more readable ones.

Other comments and suggestions

I have a couple of suggestions regarding the structure of the article to better order the information provided within the manuscript:

The paragraphs 4.1.1. Maribor Characteristics and 4.2.1. Pułtusk Characteristics should be part (embedded or as sub-paragraphs) of paragraph 4.1. Case Studies Definition and Approach. The paragraphs 4.1.3. Maribor Case Study Observations and 4.2.3. Pułtusk Case Study Observations should be part of the Discussion section, maybe integrating the information coming from these two paragraphs with the one provided for the two case studies in lines from 592 to 629.

In fact, following the instructions for authors the Results section should just “Provide a concise and precise description of the experimental results, their interpretation as well as the experimental conclusions that can be drawn”.

Following the instructions for authors, paragraph 3 should be named “Materials and Methods” and paragraph 4 should be named (just) “Results”.

As regards the English language, in general there are some mistakes within the text, especially concerning the punctuation (e.g., line 66: “By demonstrating, how DIETool works in practice, we attempt to stimulate theoretical and empirical work towards more disability inclusive societies”) and adverbs’ position (e.g. line 302: “[…] who already in the past worked closely with the respective city councils”). Please check and correct. However, in paragraph “4.2.1. PuĹ‚tusk Characteristics” there is need of more extensive proofreading as regards the English language that at this stage is inadequate for publication.

Author Response

RESPONSE TO REVIEWER 3:

We thank the reviewer for acknowledging our work. We also appreciate recommendations provided for improving the paper. The comments have been addressed one by one and the full list of amendments is provided here:

Section 1 & 2:

Lines 69-70: The sentence has been simplified as follows. “social sustainability should encapsulate questions, such as how to design socially sustainable cities concerning social inclusion also of those with diverse impairments.”

Line 73: We specified the subject of the testing as follows: “Section 4 demonstrates results of the DIETool testing within two European cities, Maribor (Slovenia) and PuĹ‚tusk (Poland).”

Line 86: “get excluded” changed to “get often excluded”

Section 2.1.: As recommended, social equity and social inclusion paradigms have been briefly explained right after their mention as follows: “Social inclusion, viewing it holistically, entails access to environment, structures, services, processes, products and information [4-8, 11, 13, 14, 31, 40], while social equity ensures that such access is fair, just and equitable to all the people (including people with disabilities). It recognizes and respects diversity and through equitable governance enables everyone to exercise their rights and duties as equal members of society [9, 10, 12, 35, 40].”

Section 3 and Section 4:

Table 1 has been replaced with the corrected typo in the word “characteristics”. Table 1: We thank the reviewer for providing an idea as of how to link the column “Characteristics…” to Appendices and specific indicators introduced therein. The table presented in the paper is a result of initial design and shows main ideas for further indicators design. It really speaks about the tool being process-related and therefore alive. In addition, the DIETool as a theoretical framework is currently under further development. Having said that, we at this stage did not decide to adapt the content in this column. However, in further stages we will take the reviewer’s suggestion into consideration as to better demonstrate the development process of the tool.

Paragraph 3.2.: Thank you for noting that we missed this important information. The clarification is provided as follows: “The overall performance score for each area is given by the normalized average score of all the indicators calculated within the respective area. The total DIPI score for the city is then calculated by the normalized average score of all the areas.”

Line 330: Closing bracket has been added.

Figure 1: We thank the reviewer for their thorough eye. We double checked the Figure and the scores, also by simulating a higher score in order to visualize the change of the straight line. The Figure is however correctly displayed. The line crosses between Area 2 and Area 17 in an almost straight line because the Area 1 scores 2,357, which is directly on “the way” between the two neighbouring areas.

Lines 362, 366, 375: In-text referencing to Figure 1 and Figure 2 has been corrected. No error detected any more.

Line 369: We have added an explanation to “explore deeper” as follows: “…the DIETool has been designed to explore deeper as of how the city performs in individual areas and individual indicators.”

Figure 2 and Figure 4 have been replaced with more readable ones without or little overlapping between indicators’ numbers. In addition, for publication purposes the in-text figures are accompanied with figures in higher resolution, uploaded into the submission system.

Other comments and suggestions:

Suggestion for the paragraph 4.1.1. & 4.2.1. to be embedded into paragraph 4.1.: As proposed, the two sections have been moved up under section 3.3. Case Studies Definition and Approach (previously mistakenly numbered as 4.1.) The new sections are now renumbered and renamed as 3.3.1. Maribor Case Study & 3.3.2. Pułtusk Case Study.

Suggestion for the paragraph 4.1.3. & 4.2.3. to be embedded into Discussion section: The two paragraphs provide only a brief conclusion in terms of observations found during the testing. We find them corresponding to the instructions for authors to provide in the Results section a “brief description of the experimental results, their interpretation as well as the experimental conclusions”. Further findings are discussed as required in the Discussion section.

Instructions for authors to the Results section: We attempt to correspond with the reviewer’s recommendation and the instructions for authors to limit the Results section to “brief description of the experimental results, their interpretation as well as the experimental conclusions”: a) by moving paragraphs 4.1.1. and 4.2.1. up under section 3.3. to limit the Results section only to results, and b) by leaving Case Study Observations (4.1.2. & 4.2.2.) as brief conclusions to the presented results under 4.1.1. & 4.2.1.

Paragraph 3 should be named “Material and Methods” & Paragraph 4 should be named “Results”: Both the Sections have been renamed accordingly.

English language: The noted mistakes have been revised. Also, the section 4.2.1. has been improved regarding English language. Additionally, full proofreading service has been requested to be incorporated into the publication process as a service for an additional charge. Therefore, please consider this comment addressed as it will be done as soon as the paper is accepted.

Reviewer 4 Report

Thank you for the opportunity to review the paper.

This study measures disability inclusion performance in two cities: Maribor (Slovenia) and Pultusk (Poland), using the Disability Inclusion Evaluation Tool. I really liked the whole paper. The authors provided a good introduction, great background, an excellent overview of DIETool with details in Appendices, a good application of the tool in two cities with different populations, and at the end discussed the pros and cons of this tool including the missing weighting system.

My few comments and suggestions are below:

Adding a study area map (figure) and two maps showing each city with areas included in the case study and excluded clearly mentioned. Roads and other layers would be helpful as well. Line 38-39: “Through extensive literature review… provide some reference here. At many places in reference, “e.g.,” or similar words are used such as Line 100, 119, 123, 590, etc. There is no need to use these words. Line 362 & Line 375: Not sure what is “Error! Reference source not found” Please explain or change the wording. Figure 4 and Figure 5 have overlapping labels. Increase the figure size or decrease the labels. Line 495: “Similarly to the previous example of the Maribor case study, we provide 5 area-specific figures also for Pultusk”. This is not correct. There are 8 insets (in Figure 2). Please use insets or similar words instead of Figures to avoid confusion and change the sentence according to the correct number of insets.

I do not have any other problem with the manuscript proceeding to publication.

Author Response

RESPONSE TO REVIEWER 4:

We are glad to hear that the reviewer liked the paper and we sincerely appreciate their recognition of our work. Hereby, we provide a list of responses to reviewer’s comments:

Adding a study map (figure): We thank the reviewer for their proposal. Indeed, we had been considering adding a map but only aiming at localizing the two cities within the European context, as both cities are less known in the eyes of global audience. Finally, we did not decide to do so because such a map is nowadays easily accessible on the web. Furthermore, responding to the reviewer’s proposal, a study area map was considered, too. However, in our view such a map does not help to explain the DIETool. The tool is not related to geographical coverage (except in some specific indicators e.g. accessible public spaces) but rather it aims at awaking the city organizational structures and relevant stakeholders to become sensitive and active towards actual accessibility in practice. The domain of such accessibility is herein seen holistically throughout city structures within the proposed 20 areas of city life (e.g. employment, education, culture), rather than just geographical coverage.

Line 38-39: Thank you for expressing the need for adding references at this occasion in the text. In our view, this sentence provides only an introduction into our work by specifying how we came to the findings listed in the subsequent bullet points. Right after listing them, we then continue drawing up the background to this list of findings by gradually introducing some of the most important studies and documents related to our work. These are 68 in total and if the reviewer would insist on incorporating them already in this sentence, we could alternatively refer to them as [1-68].

The use of e.g.: On certain occasions with the use of e.g. we wished to emphasize that the provided references are only a few examples of a more extensive corpus of works discussing that specific issue. For this reason, we decide to keep e.g. at those occasions.

Line 362 & line 375: The error was a result of improper in-text referencing to Figure 1 and Figure 2, which has now been corrected. Thank you for pointing out to this technicality.

Figures with overlapping labels have been replaced with more readable ones without or little overlapping between indicators’ numbers. In addition, for publication purposes the in-text figures are accompanied with figures in higher resolution, uploaded into the submission system.

Line 495: The sentence has been corrected as follows: “Similarly to the previous example of the Maribor case study, we provide 5 area specific insets (Figure 4) also for PuĹ‚tusk.” Also, on other occasions, when appropriate the term “inset” has been used to specify the inset figures within Figures 2 and 4.
